# Fully Incomplete Information for Multiview Clustering in Postoperative Liver Tumor Diagnoses

**DOI:** 10.3390/s25041215

**Published:** 2025-02-17

**Authors:** Siyuan Li, Xinde Li

**Affiliations:** 1School of Cyber Science and Engineering, Southeast University, Nanjing 211100, China; lisiyuan1011@163.com; 2Key Laboratory of Measurement and Control of CSE, School of Automation, Southeast University, Nanjing 210018, China; 3Southeast University Shenzhen Research Institute, Shenzhen 518063, China; 4Nanjing Center for Applied Mathematics, Nanjing 211135, China

**Keywords:** incomplete multiview clustering, feature reconstruction, fuzzy clustering, information theory

## Abstract

Multiview clustering (MVC) is a proven, effective approach to boosting the various downstream tasks given by unlabeled data. In contemporary society, domain-specific multiview data, such as multiphase postoperative liver tumor contrast-enhanced computed tomography (CECT) images, may be vulnerable to exploitation by illicit organizations or may not be comprehensively collected due to patient privacy concerns. Thus, these can be modeled as incomplete multiview clustering (IMVC) problems. Most existing IMVC methods have three issues: (1) most methods rely on paired views, which are often unavailable in clinical practice; (2) directly predicting the features of missing views may omit key features; and (3) recovered views still have subtle differences from the originals. To overcome these challenges, we proposed a novel framework named fuzzy clustering combined with information theory arithmetic based on feature reconstruction (FCITAFR). Specifically, we propose a method for reconstructing the characteristics of prevailing perspectives for each sample. Based on this, we utilized the reconstructed features to predict the missing views. Then, based on the predicted features, we used variational fuzzy c-means clustering (FCM) combined with information theory to learn the mutual information among views. The experimental results indicate the advantages of FCITAFR in comparison to state-of-the-art methods, on both in-house and external datasets, in terms of accuracy (ACC) (77.5%), normalized mutual information (NMI) (37.9%), and adjusted rand index (ARI) (29.5%).

## 1. Introduction

Liver tumors represent a significant contributor to mortality and are associated with various diseases, posing a considerable threat to human health [1,2,3]. A widely utilized method for the postoperative diagnosis of liver tumors is multiphase contrast-enhanced computed tomography (CECT) [4]. Compared with single-view methods, in clinical practice, multiphase views provide consistent and complementary information from the same patient [5,6], thus providing more precision regarding postoperative liver tumor type. Liver tumor CECT images of a patient can be divided into four phases: non-contrast (NC), arterial phase (AP), portal venous (PV), and delay phase (DP) [7,8]. Judging tumor type from CECT images is a laborious and time-wasting task for clinicians, meaning that a large number of these images are not “gold standard”. Moreover, some medical workers do not have enough experience. Hence, to further utilize such unlabeled multiview data, it is essential to introduce an unsupervised method, such as multiview clustering (MVC) [9,10,11,12]. Unfortunately, due to data theft and patient privacy, it is hard to collect complete data during transmission.

To solve the above-mentioned problems, plenty of clinical scenarios have been pro- posed with the aim of exploring information from incomplete data without labeling, which are set as incomplete multiview clustering [13,14,15] (IMVC). IMVC aims to address the issue of heterogeneous features across complete and incomplete views, where the main insight is to predict missing view features. There are two categories of IMVC methods in clinical diagnosis: traditional and deep learning IMVC. Traditional IMVC methods can be categorized as multi-kernel learning [16], matrix factorization [17,18], tensor factorization [19,20] or graph learning [21,22,23,24]. These methods are restricted in obtaining latent features, which increase computationally complex calculations. Other than traditional IMVC methods, deep IMVC methods make rich use of the latent features to achieve missing view reconstruction [25,26,27,28,29]. Although these methods present acceptable performances, the following issues remain:These methods rely significantly on complete instances (both views exist), which are often unavailable in clinical practice. For instance, in postoperative liver tumor diagnosis, few patients willingly provide complete views, making it hard to obtain complete samples.Direct feature prediction may not highlight key features; then, easily predicted error features affect the clustering performance [25,26,30,31,32], especially in the absence of paired samples.Due to noise interference, recovered missing views still have subtle difference from the originals [33,34]. With clustering tasks without labels, subtle differences may also lead to error clusters in samples.

Thus, it is hard to accomplish multiview clustering with fully incomplete information and in all instances missing at least one view.

In this study, we introduce a framework called fuzzy clustering combined with information theory arithmetic based on feature reconstruction (FCITAFR), as indicated in Figure 1. The framework primarily comprises two modules: a feature reconstruction prediction module and a fuzzy clustering contrast learning module. More specifically, we firstly reconstruct and correlate features of the existing views for all samples, aiming to optimize the latent features and enhance the relations between them. Based on the reconstructed features of views, we utilize a variational fuzzy c-means clustering (FCM) arithmetic to learn the membership among samples and then combine this with information theory [31]. We obtained the Euclidean distance among views and combined this with the membership matrix to obtain the similarity between single views. Then, we obtained the fusion similarity between different views combined with information theory, thus ascertaining consistence information among views. The major contributions of our work are as follows:Different from existing predicting methods, we reconstruct the features of existing views to enhance the relations among features.We propose a novel arithmetic that utilizes variational FCM arithmetic combined with information theory to learn the mutual information among views.The experimental results from two multiphase liver tumor datasets demonstrate that FCITAFR outperforms the state-of-the-art IMVC methods in terms of ACC (77.5%), NMI (37.9%), and ARI (29.5%).

## 2. Related Work

### 2.1. IMVC Methods

Traditional IMVC methods are categorized as multiply kernel learning, graph learning, and tensor factorization. By using the complete part of the datasets, the multiply kernel IMVC method creates the kernel matrix of the incomplete part. In order to regularize the clustering matrix, Liu et al. [35] integrated past information and gathered each imperfect base matrix produced by missing views with a clustering matrix. Graph learning-based IMVC methods gather patterns by extracting graph structure information. Chao et al. [24] proposed instance-level fusion and high-confident guiding to obtain complementary information; then, instance-level contrastive learning was utilized to obtain consistent information. Tensor factorization-based IMVC methods seek to incorporate tensor constraints to characterize the high-order correlations among tensors and elucidate the internal structure associated with cross-view data. Li et al. [36] proposed an IMVC method based on tensors to disperse multiview block-diagonal structure knowledge among views.

Deep IMVC methods, of which contrastive learning is one of the most significant, can learn the deep features of views. Xu et al. [30] proposed an adaptive feature projection-based deep IMVC method, where the auto-encoders were modeled to reconstruct missing data. Lin et al. [31] proposed a deep contrastive and dual-prediction framework to learn the consistence mutual information among different views via contrastive learning and minimized conditional entropy. Zhang et al. [29] proposed a self-attention-based encoder network and obtained common information by learning feature sub-vectors.

### 2.2. IMVC for Medical Image Analysis

In clinical practice, the numbers of paired samples are seriously limited in multiview learning. Some methods remove incomplete samples from the dataset [26]; however, removing data from medical datasets is not optimal, since incomplete samples still possess valuable information. Recently, recovering missing data is the mainstream IMVC method in medical practice. In terms of traditional methods, Wen et al. [16] proposed a framework for authenticating involuntary multiple muscle tics in children based on multiview (instance) and multiclass (clusters) features. Peng et al. [17] proposed a group sparse algorithm to combine non-negative matrix factorization and an orthogonal subspace. Different from traditional methods, deep learning methods aim to use encoders to represent the latent features of views, learning the deep relationships between them. Pan et al. [37] proposed a statement framework based on the space constraint for diagnosing brain diseases from incomplete multiview neuroimages, addressing the lack of positron emission tomography (PET) in patients with only brain MRI.

Nevertheless, these methods may have more limitations. Although the key idea of other methods has directly predicted the features of missing views, they may ignore some features and make the missing view as a wrong clustering. In order to solve this problem, the motivation is to design an arithmetic to reconstruct the features of instances, which can adequately reconstruct the features that are easy to ignore. Different from the aforementioned arithmetic, FCITAFR reconstructs the features of existing views before predicting the features of missing views. Moreover, we utilize the fuzzy clustering arithmetic combined with the information theory to learn the common features of different views after they are predicted.

## 3. Methodology

### 3.1. Notations

To facilitate the discussion, we let the number of views be 2. $X^{v}\in R^{N\times D}$ denotes the v-th view with N being the number of instances, and each has D dimensions. $Z^{v}\in R^{N\times L}$ and $L^{v}\in R^{N\times L}$ are denoted as the v-th (v∈{1, 2}) view with N being the number of samples, and L being the dimensions in latent spaces and feature reconstruction, respectively. Here, further notations and descriptions are listed in Table 1.

Based on the aforementioned definitions, we present the objective’s comprehensive loss function:(1)$$L=L_{fcit}+\lambda_{1}L_{rec}+\lambda_{2}L_{pre},\;$$
where $L_{fcit}$, $L_{rec},$ and $L_{pre}$ are fuzzy clustering with information loss, within-view reconstruction loss, and feature reconstruction prediction loss, respectively. The parameters $\lambda_{1}$ and $\lambda_{2}$ are balance parameters of loss functions; initially, we simply set them as 0.1.

### 3.2. Within-View Reconstruction

We pass all views through autoencoders to obtain the latent feature representation $Z^{v}$ by(2)$$L_{rec}=\sum_{v=1}^{V}\sum_{n=1}^{N}{\left||X_{i}^{v}-d^{v}(e^{v}(X_{i}^{v}))|\right|}_{2}^{2},\;$$
where $X_{i}^{v}$ denotes the i-th sample in $X^{v}$. Meanwhile, the latent representation in the v-th view for all instances is given by(3)$$Z_{i}^{v}=d^{v}(X_{i}^{v}).$$

### 3.3. Feature Reconstruction Prediction

Based on the latent representation $Z^{v}$ obtained by $L_{rec}$, we propose a methodology for reconstructing the characteristics of samples, seeking to take into account the interrelationships between features and among various samples within a singular perspective. The primary objective is to determine the weights of all features in accordance with the specified indicators [38]. When acquiring latent representations from each perspective, the latent spaces encompass various samples and distinct characteristics. More specific, $Z_{ij}^{v}$ consists of n samples, with each sample characterized by m features. The i-th sample indicates $Z_{i,:}$, and the j-th feature indicates $Z_{:,j}$. We employ the entropy weighting method to analyze the obtained latent features, associating them with each unique characteristic of every sample, and subsequently generate a novel latent representation.

Firstly, for the feature of each sample, we decline the difference of the distribution. We gathered the average of features for all samples, and obtained the distribution matrix $Y_{i,:}\in R^{N\times L}$ as follows:(4)$$Y_{ij}=\frac{Z_{ij}}{\bar{Z_{i,:}}}\;\left(i\in \left[1,n\right],j\in \left[1,m\right]\right),$$
where $\bar{Z_{i,:}}$ donates the average of features for all samples.

Next, we analyzed the relationships between features across various samples, specifically focusing on the j-th column of Y. We computed the entropy value $E_{:,j}$ to integrate the different features:(5)$$E_{:,j}=\frac{1}{n}\sum_{i=1}^{n}p_{ij}logp_{ij},$$
where $p_{ij}=\frac{Y_{ij}}{\bar{Y_{:,j}}}$, and $\bar{Y_{:,j}}$ donates the averages of all samples for all corresponding features.

In conclusion, it is essential to calculate the entropy coefficient $W_{j}$ for each column and subsequently multiply the aggregate coefficients by the initial latent representation $Z^{v}$. The formulas for these calculations are as follows:(6)$$W_{j}=\frac{1-E_{:,j}}{m-\sum_{j=1}^{m}E_{:,j}};\;$$(7)$$L^{v}=\sum_{j=1}^{m}W_{j}Z^{v}$$

After obtaining the reconstructed latent representation $L^{v}$, we utilized generators to predict the missing views. Our main task was to connect all features of each sample. In order to predict the features of the other view, the importance is not to ignore the key features. The loss function $L_{pre}$ can be obtained as follows:(8)$$L_{pre}={\left||G^{12}(L^{1})-L^{2}|\right|}_{2}^{2}+{\left||G^{21}(L^{2})-L^{1}|\right|}_{2}^{2},$$
where $G^{12}$ and $G^{21}$ are generators with a multilayer, fully-connected network.

### 3.4. Fuzzy Clustering with Information Theory

To learn more mutual information among views, we used variational fuzzy c-means clustering (FCM) [39], combined with information theory [31,32], to obtain the relationship among samples and obtain the fusion similarity between views. We used all samples as the clustering centers to determine the degree of membership for the other samples. First, we obtained the similarity matrix $S^{v}\in R^{N\times N}$ of single views, which is formulated as follows:(9)$$S^{v}=\frac{1}{V}\sum_{i=1}^{n}\sum_{j=1}^{n}{\left(U_{ij}^{v}\right)}^{2}{\left||l_{i}-l_{j}|\right|}^{2}\;(v\in \{1,2\}),$$

In this function, $S^{v}\in R^{N\times N}$ donates the time complexity matrix of the algorithm, which is calculated by $L^{v}{\left(L^{v}\right)}^{T}$ for each view. $L^{v}\in R^{N\times L}$ donates the feature matrix of each view after feature reconstruction, which consists of N number of samples and, each sample has L features. In order to determine the similarity among all samples in each view, we calculated the feature matrix ($R^{N\times L}$) times the transposed feature matrix ($R^{L\times N}$). As a result, we can obtain the time complexity matrix. And ${\left||l_{i}-l_{j}|\right|}^{2}$ denotes the Euclidean distance between the i-th and j-th samples (i≠j), and l are variational samples. $U_{ij}^{v}\in R^{N\times N}$ donates the membership matrix, which stands for the degree of membership among samples; a larger number means more similarity. Then, the membership matrix among views is calculated as follows:(10)$$U_{ij}^{v}={\left(\frac{1}{m{\left||l_{i}-l_{j}|\right|}^{2}}\right)}^{\frac{1}{m-1}},\;$$
where m represents the membership factor, which is set as 2 in this study.

Subsequently, we enhanced the mutual information between $S^{v}$ by integrating it with information theory. The final layer of the encoder, which is based on the softmax function [33], can be understood as representing a probability distribution that facilitates clustering. Thus, the cross-view joint probability distribution matrix $S_{ll\prime }\in R^{N\times N}$ is calculated by $S^{1}{\left(S^{2}\right)}^{T}$. Let $S_{l}$ and $S_{l\prime }$ become the marginal probability distributions, which denote the l-th row and the $l^{\prime }$-th column on $S_{ll\prime }$. The function $L_{fcit}$ can be represented as follows:(11)$$L_{fcit}=-\sum_{l=1}^{L}\sum_{l\prime =1}^{L}\frac{S_{ll\prime }}{S_{l}^{\alpha +1}\cdot S_{l\prime }^{\alpha +1}},\;$$
where α donates the entropy balancing parameter, which is set to 9. The implementation details of FCITAFR are indicated in Algorithm 1.
**Algorithm 1** Fuzzy clustering combined with information theory arithmetic based on feature reconstruction (FCITAFR)**Input:** Incomplete dataset {Xiv}.**Output:** Total loss L.1:  Initialize overall loss L, epoch number is E, the number of clusters K; the parameters λ1, λ2.2:  **for**
epoch≤E **do**3:         Compute with-in view reconstruction loss by (2).4:         Compute feature reconstruct prediction loss by (8).5:         Compute fuzzy clustering with information theory loss by (11).6:         Compute total loss by (1) and update balance parameters.7:         E=E+1.8:    **return**
L9:    Update parameter λ1.10: Update parameter λ2.

## 4. Experiments

### 4.1. Experimental Setting

#### 4.1.1. Dataset

We performed experiments using both our internal dataset and external multiview liver tumor CECT dataset, with all volumes obtained from Philips iCT 256 scanners using NC and AP imaging.

Zhongda includes 82 patients with 328 multiphase CT scans obtained from Zhongda Hospital, Southeast University. Each volume has an in-plane dimension of 512 × 512, with spacing varying between 0.601 mm and 0.851 mm. The number of slices in the volumes varies from 36 to 139, with a slice spacing of 2.5 mm.Zheyi includes 475 patients with 1900 multiphase CT scans obtained from The First Affiliated Hospital of Zhejiang University. Each volume has an in-plane dimension of 512 × 512, with spacing varying between 0.560 mm and 0.847 mm. The number of slices in the volumes varies from 25 to 89, with a slice spacing of 3.0 mm.

To ensure clustering validity, we retain 55 and 286 instances (for Zhongda and Zheyi, respectively) after removing samples with tiny tumors. These instances are annotated with two levels (M0 and M1) of microvascular invasion (MVI), which is a popular indicator of postoperative liver diagnosis for hepatocellular carcinoma (HCC) [40]. As shown in Figure 2, for the CECT images that have five instances, and each sample only has one view, which means all patients are missing one view. Here, we introduce the phases of NC and AP as the views for all the methods, with missing rates of {10%, 30%, 50%, 70%, 100%} used to validate the methods in incomplete multiview scenarios.

#### 4.1.2. Competitor

We compare FCITAFR with the state-of-the-art methods as follows:SNFR [29] considers feature reconstruction to learn common information, utilizing information theory to learn the mutual information based on the reconstructed feature.COMPLETER [31] learns the common information and constructs a contrastive prediction module to infer missing views.DCP [32] extends more than two views based on [31] and divides these into core-view and complete-view algorithms.Prolmp [33] is a dual contrastive learning-based IMVC that learns the relationship between the view-level and instance prototypes.SURE [34] considers the view-unaligned and missing problems.

#### 4.1.3. Implementation Details and Metrics

We implement our approach using PyTorch 1.8.0 on an NVIDIA GeForce RTX 2080Ti GPU (NVIDIA, Santa Clara, CA, USA). The Adam optimizer is employed, with an initial learning rate of 0.0001 applied across all datasets. Initially, we set the batch size to 256 and the training epoch to 500 for all datasets. We set up the two trade-off hyper-parameters $\lambda_{1}$ and $\lambda_{2}$ to 0.1 and 0.1, respectively. The evaluation metrics validate the cluster performances as follows: ACC, NMI, and ARI.

### 4.2. Comparison with State-of-the-Art

Table 2 indicates the clustering results of all methods with the different missing rates {10% ,30%, 50%, 70%} and compares with DCP under a missing rate of 100%. This can explain the clustering performances in different missing rates and highlights better clustering performances in high missing rates compare with competitors. High missing rates are more common in clinical practices, and a robust framework can suit them, where the framework can obtain a better clustering performance in high missing rates, called robust framework. Note that “-” means that the algorithm cannot be operated smoothly due an to out-of-memory error.

Based on the statistical analysis of the results, the generated observation can be summarized as follows:(1)Compared with other methods, FCITAFR clustering performances can achieve the best results. For instance, in the low missing rates, such as 10%, FCITAFR achieves the best clustering performances on all datasets, where ACC improved 6.1% and 5.7% compared to the second-best method on Zhongda and Zheyi, respectively. Moreover, NMI improved obviously on Zhongda, which improved about 11% compared to the second-best one, and also improved about 3% on Zheyi. Furthermore, ARI improved about 2% and 9% on Zhongda and Zheyi, respectively. On the missing rate of 30%, mostly clustering performances obtained the best one on FCITAFR, especially on Zhongda, which improved about 10% to 20% of all clustering performances compared to the second-best one.(2)FCITAFR has significant advantages. All other deep learning methods, SNFR, COMPLETER, DCP, Prolmp and SURE, directly predict the features of missing views and may predict incorrect features, even with lower missing rates. FCITAFR further reconstructs the features of latent representations of existing views, which obtain the average of all features, and then utilizes the average to recalculate the original features. This informs the obvious efficacy of further reconstruction for the features of latent space.(3)With the increasing missing rate, although the clustering performance of FCITAFR declined, it is still outstanding compared to competitors. For example, on the missing rate of 50%, although some clustering performances obtained the second-best one, ACC improved about 1% on all datasets compared to the second-best method. On the other hand, with the missing rate of 70%, the clustering performances improved about 1% to 2% on Zheyi, and NMI improved about 7.5 on Zhongda compared to the second-best one. Our method suits the case of all instances missing one view. We compared with DCP with the missing rate of 100%, which improved about 2% for all performances compared to DCP.

### 4.3. Time Cost Comparison

The running time comparison between the Prolmp [33] and SURE [34] are indicated in Table 3. As shown in Table 3, our method is remarkably efficient compared to Prolmp and SURE on all datasets with the missing rate of 0.5. It may save about 10 s on Zhongda and save 1 min on Zheyi. Note that all methods run on the GPU.

### 4.4. Parameter Analysis

To analyze the impact of trade-off hyper-parameters, we conducted a series of experiments with the missing rate set as 0.3 on Zhongda. $\lambda_{1}$ and $\lambda_{2}$ are set to intervals of {0.01, 0.1, 1, 10}. As shown in Figure 3, based on the statistical analysis, the following observations can be summarized:
(1)When it is fixed, the performance will be worse when it is too large. For instance, when we set $\lambda_{2}$ as 0.01, our method can obtain the best performances when $\lambda_{1}$ fixed as all figures. However, when the parameter $\lambda_{2}$ is increased, the clustering performance significantly declined when it set as 0.1, which declined about 3%, 5% and 6% of ACC, NMI and ARI, respectively. And when $\lambda_{2}$ is set as larger, the clustering performances continually declined. This is because the feature reconstruction prediction loss is used, as predicted by the latent features, and when it is set too large, the structures of features will change more, where original features may be destroyed.(2)When $\lambda_{2}$ is fixed, the performance will decline significantly when $\lambda_{1}$ is too large or small. When we set $\lambda_{1}$ as 0.1, our method can obtain the best performances when $\lambda_{2}$ fixed as 0.01 and 0.1. When $\lambda_{1}$ set as 0.01, although ACC moderately decreased, NMI and ARI decreased about 7% and 8%, respectively. On the other hand, when $\lambda_{2}$ is fixed as 0.01, all clustering performances frequently declined. Moreover, when $\lambda_{2}$ is fixed as other figures, $\lambda_{1}$ fluctuates in different sets. This is because a small and a large trade-off parameter may destroy the robustness of the framework, which can cause the features to misregister the original features.(3)When both $\lambda_{1}$ and $\lambda_{2}$ are less than 0.1, the clustering performances will be worse when $\lambda_{1}$ is 0.1. On the other hand, when both $\lambda_{1}$ and $\lambda_{2}$ are too large, the clustering performances will significantly decline.

Therefore, after analyzing the different sets of trade-off hyper-parameters, the value of $\lambda_{1}$ and $\lambda_{2}$ are suggested as {0.1} and {0.01}, respectively.

### 4.5. Ablation Studies

To further confirm the effectiveness of our approach, we performed ablation experiments on Zhongda with a missing rate of 0.3, as indicated in Table 4. A series of FCITAFR variants is modeled with different combinations of $L_{fcit}$, $L_{rec}$ and $L_{pre}$. We can observe that the performance declined sightly when one or two loss functions ablated; all loss functions play significant roles in the arithmetic. The statistical analysis can be shown as follows:
(1)The fuzzy clustering with information theory loss $L_{fcit}$ may be the most important loss function in the method. In the first line, with only $L_{fcit}$, the clustering performances will decline by only about 1%. When this function is ablated, the clustering performance ACC is declined by about 3% to 6%. Such as in line 3, ACC, NMI, and ARI decreased about 6%, 4% and 10%, respectively, when $L_{rec}$ is also ablated. This is because it will hardly ignore the significant features for each sample when this loss function is ablated, and $L_{rec}$ also can learn the important features of views.(2)For with-in view reconstruction loss $L_{rec}$, many works demonstrate that autoencoders are widely used in unsupervised learning; autoencoder structure is helpful to avoid the trivial solution. Other loss function also needed to learn the latent representation from encoders by autoencoders. Overall, the aforementioned observations have verified the effectiveness of the proposed loss function in our method. In the third and the fifth line, the clustering performances declined about 2% to 5% when $L_{rec}$ was ablated.(3)For feature reconstruction prediction loss $L_{pre}$, the feature reconstruction can change the features of views, which may improve the relationship by different features, and avoid ignoring some important features and margin features. Therefore, this loss function may improve the clustering performances. In addition, in the fourth line, ACC declined about 2% when $L_{pre}$ was ablated, and NMI and ARI decrease more than 10% without $L_{pre}$. Furthermore, compared with line 4, ACC, NMI, and ARI are increased about 3%, 10%, and 7%, and it can be clearly seen that reconstructing the features of existing views is so significant. Obtaining the average of features about existing views can let the features equalize more, which can close the distance of different features.


Overall, the fuzzy clustering with information theory loss $L_{fcit}$ is the most significant loss function in this method, and $L_{fcit}$ $L_{rec}$ have mutual promotion and also can cause a minor effect when only $L_{fcit}$ exists. And the other two loss functions also play an important role in this algorithm, since they may obtain a better clustering performance when they exist together than only exist alone.

### 4.6. Convergence Analysis

Furthermore, we visualized the latent space via t-SNE on Zheyi with the missing rate of 0.3, as shown in Figure 4. Based on the statistical analysis, the following observations can be summarized: (1) The clustering performance sharply increased during the first 75 epochs; this is because FCITAFR effectively learns the structures of the framework and then the features learn adequately. As a result, the clustering performance levels stay consistent. These findings indicate that the proposed method achieved training convergence. (2) The loss is significantly decreased during the first 75 epochs. This is because FCITAFR utilizes the latent features and reduces the gap between the original features and predicted features.

To explore the convergence procedure of FCITAFR, we plotted the values of ACC, NMI, and ARI, and the total loss on Zhongda with the missing rate of 0.3 in the training stage, as shown in Figure 5. In comparison with the qualitative results of other approaches, our method shows an improved performance with more epochs, leading to a greater distinction between the features of the two different types.

## 5. Conclusions

In this paper, we propose a novel incomplete multiview clustering (IMVC) method with fully incomplete information. First, we built a feature reconstruction arithmetic to reconstruct the features of existing views, utilizing them to predict missing views. Then, we combined information with variational fuzzy c-means (FCM) clustering to learn the mutual information among views. The experimental findings show that FCITAFR outperforms the leading methods on both the internal and external datasets. Also, we analyzed the impact of trade-off hyper-parameters, and analyzed the effect of one parameter and both two parameters. Although our arithmetic obtained an outcome performance compared with one that was state-of-the-art, it only suited the multiview scene of two views, where multiview datasets in clinical experiments included more than two views; our method only can analyze the common and complementary information two-by-two. On the other hand, although our method is a time-saving arithmetic compared with the competitors during the time cost experiment, our method is not suited to the situation of more than two views; it is also waste of time, since the experiments need to analyze the views two-by-two. In the future, we will extend these FCITAFR insights to unaligned samples and extend the framework to the situation of more than two views, thus enhancing the capabilities of FCITAFR in real clinical scenarios.

## Figures and Tables

**Figure 1 sensors-25-01215-f001:**
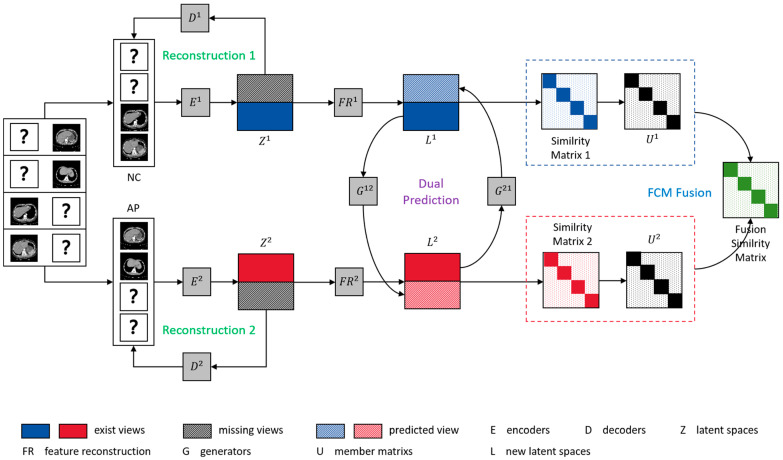
The framework of FCITAFR, where “?” donates the missing views. The within-view reconstruction loss $L_{rec}$ uses encoders to obtain the latent representation $Z^{v}$. Then, via feature reconstruct prediction loss $L_{pre}$, in order to reconstruct the latent representation of existing views to obtain new latent representation $L^{v}$, the generators *G* predict the missing views of each sample. Finally, via fuzzy clustering with information theory loss $L_{fcit}$, in order to obtain the membership matrix $U^{v}$ of each view, the fusion similarity matrix among views can be obtained via information theory.

**Figure 2 sensors-25-01215-f002:**
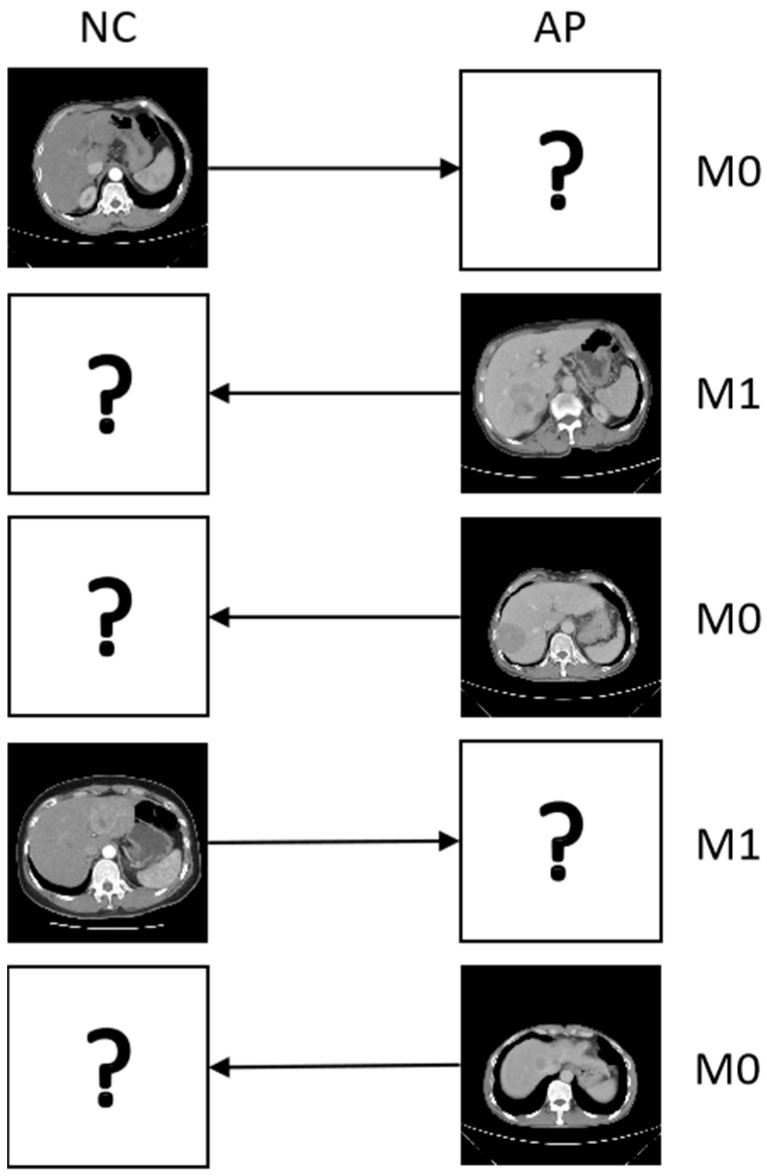
Five instances of liver tumor CECT images from the different periods; “?” denotes a missing view and “→” denotes mapping the missing view.

**Figure 3 sensors-25-01215-f003:**
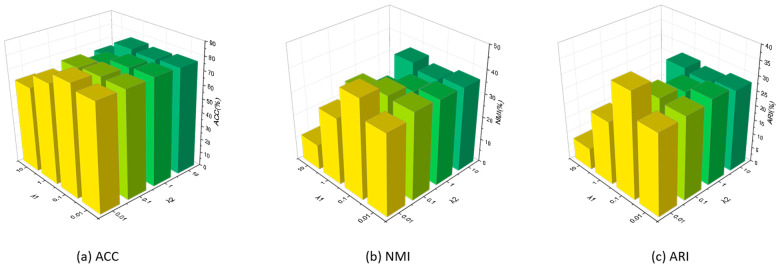
Parameter evaluation on Zhongda with a missing rate of 0.3.

**Figure 4 sensors-25-01215-f004:**
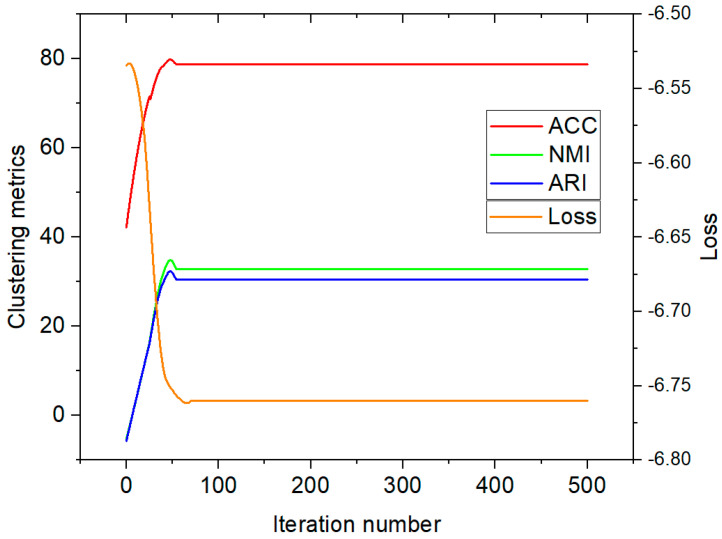
Visualization of latent space via t-SNE on Zheyi with the missing rate of 0.3. With an epoch of 500, compared with other methods, FCITAFR constructed a more discriminative latent space.

**Figure 5 sensors-25-01215-f005:**
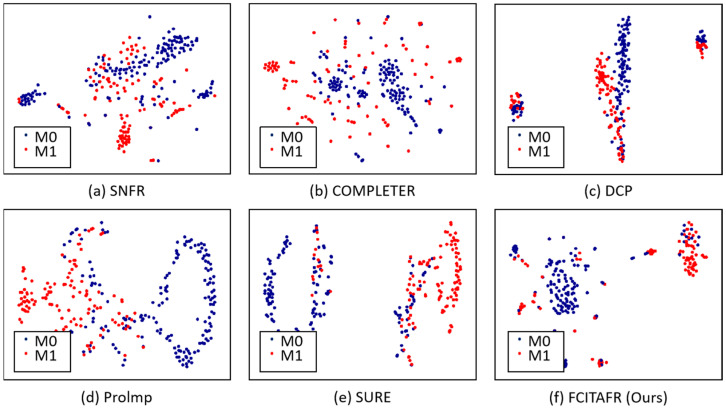
Convergence evaluation on Zhongda with a missing rate of 0.3.

**Table 1 sensors-25-01215-t001:** Notations with corresponding descriptions.

Notation	Descriptions
i,j,l,m,n,d	the notations utilized for indexing
N	the number of samples in the multiview dataset
V	the number of views in the multiview dataset
$X^{v}\in R^{N\times D}$	the data matrix corresponding to the v-th view
$Z^{v}\in R^{N\times L}$	the latent representation of the v-th view
$L^{v}\in R^{N\times L}$	the new latent representation of the v-th view after feature reconstruction
D	the dimension of $X^{v}$
L	the dimension of $Z^{v}$ and $L^{v}$
$\lambda_{1},\;\lambda_{2}$	the trade-off parameters for loss functions
$e^{v},\;d^{v}$	the encoder and decoder parameters of the v-th view
$S^{v}\in R^{N\times N}$	the similarity matrix in the v-th view
$U^{v}\in R^{N\times N}$	the membership matrix of the v-th view

**Table 2 sensors-25-01215-t002:** A clustering performance comparison of two liver tumor datasets. The 1st/2nd best results are in **bold** and underlined.

			Zhongda			Zheyi	
Missing Rates	Method	ACC (%)	NMI (%)	ARI (%)	ACC (%)	NMI (%)	ARI (%)
10% Missing	SNFR (2024)	58.2	7.0	1.7	59.1	1.2	3.3
	COMPLETER (2021)	58.2	5.9	1.8	60.8	1.7	3.5
	DCP (2022)	53.8	4.4	0.7	59.0	7.9	0.2
	Prolmp (2023)	69.5	22.3	23.2	65.1	14.1	13.6
	SURE (2022)	63.7	9.5	6.2	73.9	27.9	25.8
	FCITAFR (Ours)	**75.6**	**33.5**	**25.3**	**79.6**	**31.0**	**34.6**
30% Missing	SNFR (2024)	58.6	7.6	2.4	56.0	7.0	3.8
	COMPLETER (2021)	56.0	1.1	0.5	59.4	2.8	0.7
	DCP (2022)	56.7	3.2	0.7	57.5	0.2	0.5
	Prolmp (2023)	65.1	12.5	10.8	70.2	**23.0**	19.5
	SURE (2022)	67.3	9.4	10.5	68.9	11.6	14.0
	FCITAFR (Ours)	**77.5**	**37.9**	**29.5**	**71.0**	20.3	**19.9**
50% Missing	SNFR (2024)	57.8	10.1	1.6	59.1	7.9	4.3
	COMPLETER (2021)	61.8	10.6	7.3	60.6	3.4	1.5
	DCP (2022)	56.7	2.0	1.0	54.6	0.5	0.6
	Prolmp (2023)	76.4	29.8	**30.0**	70.7	**25.2**	**20.9**
	SURE (2022)	-	-	-	58.4	0.6	1.7
	FCITAFR (Ours)	**76.7**	**38.5**	28.2	**71.0**	22.7	15.9
70% Missing	SNFR (2024)	59.6	14.4	3.2	54.1	3.9	1.6
	COMPLETER (2021)	70.9	25.0	19.7	62.2	7.5	4.0
	DCP (2022)	56.3	1.5	0.3	54.2	0.5	0.6
	Prolmp (2023)	**73.5**	24.0	**22.5**	58.7	6.5	4.6
	SURE (2022)	-	-	-	60.8	1.7	3.5
	FCITAFR (Ours)	72.4	**32.6**	20.0	**63.6**	**10.3**	**6.3**
100% Missing	DCP (2022)	56.3	1.3	0.1	53.2	0.4	0.2
	FCITAFR (Ours)	**58.6**	**2.4**	**1.8**	**55.0**	**4.4**	**1.6**

**Table 3 sensors-25-01215-t003:** Time cost comparison.

Dataset	Method	Training Time (seconds)
Zhongda	Prolmp	75
	SURE	65
	FCITAFR (Our)	58
Zheyi	Prolmp	210
	SURE	195
	FCITAFR (Our)	155

**Table 4 sensors-25-01215-t004:** Effect of different loss functions on Zhongda with a missing rate of 0.3, “√” donates the loss function is participated in experiments.

$L_{fcit}$	$L_{rec}$	$L_{pre}$	ACC (%)	NMI (%)	ARI (%)
√			76.4	34.6	26.7
	√		76.4	37.1	29.3
		√	70.9	30.3	16.2
√	√		74.5	27.7	22.8
√		√	75.6	35.4	25.5
	√	√	74.2	31.8	22.7
√	√	√	77.5	37.9	29.5

## Data Availability

The raw data supporting the conclusions of this article will be made available by the authors on request.

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
