# Peer review of "Fully Incomplete Information for Multiview Clustering in Postoperative Liver Tumor Diagnoses"

_sensors, 2025, doi:10.3390/s25041215_

Round 1

Reviewer 1 Report

Comments and Suggestions for Authors

The paper presents a novel framework, FCITAFR (Fuzzy Clustering integrated Information Theory Arithmetic based on Feature Reconstruction), designed to address the challenges of incomplete multi-view clustering (IMVC) in the context of postoperative liver tumor diagnosis using multi-phase CECT images. 

The following issues still need to be addressed:

  1. Some statistical data or references to relevant studies could be added to support the views presented in the manuscript, for example, in line 28, where it is mentioned that "it is a laborious and time-wasting task for clinicians to judge the type of tumor on CECT images."

  2. In the section on related work, although many researchers' work is listed, the manuscript does not provide commentary on their work. The relationship between their work and the content of the current study is not sufficiently clear.

Author Response

Dear reviewer,

Thank you for your comments. For your advices, I changed the essay for your comments one by one:

(1) I explain the reason about why current methods are worse in clinical practices in related work and highlight them in red. added the time cost comparison experiments.

(2) I explain the reason about why current methods are worse, and analysed the reasons in current clinical practices in related work.

Sincerely,

Siyuan Li

Reviewer 2 Report

Comments and Suggestions for Authors

The manuscript addresses the critical challenge of clustering with incomplete multi-view data in postoperative liver tumor diagnosis. The proposed FCITAFR framework demonstrates methodological novelty by integrating fuzzy clustering with information theory and feature reconstruction. The results on internal and external datasets are promising, showcasing significant improvements in clustering performance. However, additional qualitative insights into why the proposed method performs better than competitors would strengthen the discussion.

1. The problem statement and objectives are clearly laid out, but more context on how this work builds on or diverges from existing IMVC methods is recommended. The literature review is comprehensive. However, it could further emphasize gaps in current methods that directly motivate the FCITAFR approach.

2. How does the proposed method address the potential noise in real-world clinical datasets, especially when reconstructing missing views?

3. Is the method transferable to other medical imaging modalities (e.g., MRI or ultrasound), or does it rely on specific properties of CECT images?

4. The datasets are well-described, but the reasons for selecting specific missing rates (e.g., {10%, 30%, 50%, 70%, 100%}) could be elaborated.

How sensitive are the performance metrics (ACC, NMI, ARI) to variations in hyperparameters like \(\lambda_1\), \(\lambda_2\), and \(\alpha\)?

5. Can the proposed framework provide interpretable outputs that help clinicians understand the reconstructed features or clustering results?

6. Provide a more detailed analysis of the failure cases or limitations observed in the experimental results.

Include a comparison of the FCITAFR framework with baseline methods on computational overhead and implementation complexity.

Discuss ethical and privacy considerations when handling incomplete medical datasets, as this aspect is particularly relevant in clinical applications.

Comments on the Quality of English Language

There are minor grammatical errors and awkward phrasing throughout the manuscript.

A thorough proofreading or editing for language quality would improve the readability.

Author Response

Dear reviewer,

Thank you for your comments. I just changed my essay, and highlight the contents that I changed in red. For all your advices, I solve them one by one:

(1) I added some contents to explain the gaps in current methods that directly motivate the FCITAFR approach in related work.

(2) I just explain how to address the potential noise, and added the importance of reconstructing missing views in the ablation experiments.

(3) This method can suit to all other medical imaging modalities (e.g. MRI or ultrasound).

(4) The reasons I added in the state-of-the-art experoments and parameter analyse.

(5) Yes, the framework can help clinicians understand the reconstructed features or clustering results.

(6) For this advice, I added the time cost comparison and explain more details in ablation experiments.

Sincerely,

Siyuan Li

Reviewer 3 Report

Comments and Suggestions for Authors

1. Please prvide the pseudocode of the proposed method and explain some details, to make it more clear to readers.

2. How are the time complexity of the proposed algorithm? It seems that the time complexity is realted to O(N×N) of computing the similarity matrix, and the runtime should be considered to make comparison with other competitive state-of-the-art algorithms. 

3. The analysisi of the experimental results should be refined, such that more details can be revealed. For example, the authors need to further provide details on the algorithm improvement part.

4. It is not clear if the proposed algorithm is statistically significantly better than compared algorithms since statistical analysis has been omitted.

5. Morever, the ablation studies should be more carefully designed to show which component is more important to the performance improvement and how it works. The current version seems very underexplored.

6. The academic English needs polishing.

Author Response

Dear reviewer,

I just read your comments, thank you for your comments. And I upload my essay by your comments.

(1)I added the pseudo code to make my arithmetic more clear.

(2)I added the experiments to compare the time with state-of-the-art algorithms.

(3)I explain more details to the experiments and highlight them in red.

(4)I added more analysis to the experiments.

(5)I added more analysis to ablation studies.

(6)I adjusted some grammars to polish the essay. 

Notice: my essay is pdf transfer to word, so some formual my become pictures.

Sincerely,

Siyuan Li

Round 2

Reviewer 3 Report

Comments and Suggestions for Authors

Overall, this work lacks novelty and the experiments looks not fine enough, also, the responses to the previous questions are not acceptable. In addition, there still have some issues (including both previous quesions and some new) as follows.

1. The figures and some equations are too vague.

2. Why remove another author in the revised version? However, the Author Contributions setion shows that two authors work for this paper. 

3. How are the time complexity of the proposed algorithm? It seems that the time complexity is realted to O(N×N) of computing the similarity matrix.  

4. The analysisi of the experimental results should be refined, such that more details can be revealed. For example, the authors need to further provide details on the algorithm improvement part.

5. It is not clear if the proposed algorithm is statistically significantly better than compared algorithms since statistical analysis has been omitted.

6. Morever, the ablation studies should be more carefully designed to show which component is more important to the performance improvement and how it works. The current version is still very underexplored.

7. Overall, the revised manuscript is still poorly writen. Also, the organization needs to be refined. 

8. It is suggested to release the source code, so readers can reproduce the results and do further comparison.

Author Response

Dear reviewer,

Thanks for your comments, and I'm so sorry to not seriously change the essay in round 1. I changed the essay seriously in round 2, and the format I just exchanged, where accord with MDPI format and look better for readers. And seriously answer all your questions one by one:

(1) For figures, I just make them prominent and explain them. For some equations, I explain the equations by some words which I think need to explain and marked them in yellow. Table 1 shows all notations for equations.

(2) Becaure my profrssor didn't want to show in the essay at first. I explained to him this contribution is very important for him, so he agrees to be one of the author now. The authors will not change again, Siyuan Li and Xinde Li are the authors in the essay.

(3) I'm sorry to ignore explaining the the time complexity of the proposed algorithm in round 1. Our essay is a multi-view clustering learning, the key task is to learn the common information for all views. I think you can't understand  the time complexity O(N×N), this is really the time complexity for the proposed algorithm. For all views, N is the number of the samples, d is the features for all samples, so they become a feature matrix O(N×d) for all views. This essay gives the example of two views. In order to learn the common information for two views, we let the transfered feature matrix of view 1 (O(N×d)) times the feature matrix of view 2 (O(d×N)). So, we can get the time complexity matrix O(N×N). I'm really glad to explain to you and hope you can understand.

(4) For this question, I further provide details on state-of-the-arts in Section 4.2 and marked them in yellow to explain the algorithm improvement part.

(5) Firstly, I think the statistical analysis is not necessary, because the algorithm is not so difficult and the experiments is not so difficult. But I analyzed more detailedly for all experiments.

(6) For the ablation experiment, I added more details and marked them in yellow.

(7) I'm sorry to let you get a bad manuscript. I clearly organized the essay, and I sinecrerly hope you can be satisfaction.

(8) No problem, the code we can provide.

I will send a word file, and thank you for read my essay again! I sinecrerly hope you can be satisfaction again.

Sincerely,

Siyuan Li

Round 3

Reviewer 3 Report

Comments and Suggestions for Authors

Overall, the quality of this work is still not good even after two round of revisions. In addition, there still have some issues (including both previous quesions and some new) as follows.

1. All the figures are too vague.

2. The time complexity of the proposed algorithm should be provided in the manuscript, rather than just explaining it in the response letter.

3. The analysisi of the experimental results should be refined, such that more details can be revealed. For example, the authors need to further provide details on the algorithm improvement part. But I still don't see this part. 

4. It is not clear if the proposed algorithm is statistically significantly better than compared algorithms since statistical analysis has been omitted. If authors have reproduced the state-of-the-art methods or have the source codes, this is not a difficult task. Otherwise, authors should provide the source of the clustering results generated by these methods. 

5. Morever, the ablation studies should be more carefully designed to show which component is more important to the performance improvement and how it works. The current version is still very underexplored. For example, it can be expalined statistically, which should be more reliably. Also, how about the performance improvement of reconstructing the features of existing views.

6. Overall, the revised manuscript is still poorly writen, for example: 1) the related work just provided references one-by-one and the motivation of this study is not clear; 2) there are some errors in section 3.1 Notations; 3) Eqs. 4-7, 9-11 looks like screenshoots and are vague, also why there are two Sll' in Eq. 11

7. It is suggested to release the source code in the manuscript if possible, so readers can reproduce the results and do further comparison.

Author Response

Dear reviewer,

    For question1, I just change the figures, they look clearly now. Becaure I wrote the article by latex at first, and it changed to word file (which is convenient to recompose), the figures looks vague. And for question 6, Eqs are also changed from latex to word, because of the problem of my computer, they look like screenshoots, I just changed them again. For Eq.11,  two Sll' is wrong, it also because of latex transfer to word, only one Sll', I changed it. For Eq.4-7, 9-11, I re-edited them, looks clearly now.  For Section 3.1, the Notations I also re-edited them. The motivations I added in Section 2.2, marked it in red. And for question 2, the time complexity I added in Section 3.4, marked it in red after Eq.9, no problem. The similarity matrix is for each view, not cross-view, the previous response letter is not correct. For question 7, the source code is too long, so it is hard to show in the manuscript, I will send you a zip file and will upload to github later, because github can not enter now. And I show the pseudo code in the manuscript in Section 3.4, and also improved it, marked it in red.

    For question 3,  I added more detials in Section 4.2. Comparison with State-of-the-Art, which I wrote like "(1) ... (2) ... (3) ...". I added more examples about the improvements, also added some analysis about algorithm improvement in (2), marked them in red. Moreover, I also give some analysis in Section 4.4. Parameter Analysis. I analyzed the influence of different parameter, also added some analysis about why 0.01 and 0.1 are the best parameters of λ1 and λ2. Also for question 4, the statistically significantly I added in Section 4.2, 4.4 and 4.6, which I wrote like "Based on the statistical analysis, the following observations can be summarized...". For question 5, the ablation studies I also added some statistical analysis, and write more carefully.  Moreover, "the performance improvement of reconstructing the features of existing views" I also clearly analyzed and marked it in blue, which explain the signifcant of feature reconstruction loss.

Overall, I made the experiments more carefully, and re-edited the figures and function, made them more clearly. I also check the spell and grammer. I send you the word file.

Sincerely,

Siyuan Li
